# Synthesis of ordered carbonaceous frameworks from organic crystals

Hirotomo Nishihara [1,2], Tetsuya Hirota[1], Kenta Matsuura[1], Mao Ohwada[1], Norihisa Hoshino[1],
Tomoyuki Akutagawa[1], Takeshi Higuchi[1], Hiroshi Jinnai[1], Yoshitaka Koseki[1], Hitoshi Kasai[1], Yoshiaki Matsuo[3],
Jun Maruyama[4], Yuichiro Hayasaka[5], Hisashi Konaka[6], Yasuhiro Yamada[7], Shingi Yamaguchi[8],
Kazuhide Kamiya[2,9], Takuya Kamimura[10], Hirofumi Nobukuni[10] & Fumito Tani[10]

Despite recent advances in the carbonization of organic crystalline solids like metal-organic frameworks or supramolecular frameworks, it has been challenging to convert crystalline organic solids into ordered carbonaceous frameworks. Herein, we report a route to attaining such ordered frameworks via the carbonization of an organic crystal of a Ni-containing cyclic porphyrin dimer ($Ni_2$-$CPD_{Py}$). This dimer comprises two Ni–porphyrins linked by two butadiyne (diacetylene) moieties through phenyl groups. The $Ni_2$-$CPD_{Py}$ crystal is thermally converted into a crystalline covalent-organic framework at 581 K and is further converted into ordered carbonaceous frameworks equipped with electrical conductivity by subsequent carbonization at 873–1073 K. In addition, the porphyrin's Ni–$N_4$ unit is also well retained and embedded in the final framework. The resulting ordered carbonaceous frameworks exhibit an intermediate structure, between organic-based frameworks and carbon materials, with advantageous electrocatalysis. This principle enables the chemical molecular-level structural design of three-dimensional carbonaceous frameworks.

[1] Institute of Multidisciplinary Research for Advanced Materials, Tohoku University, 2–1–1 Katahira, Aoba, Sendai 980–8577, Japan. [2] PRESTO, the Japan Science and Technology Agency (JST), 4-1-8 Honcho, Kawaguchi 332-0012, Japan. [3] Department of Materials Science and Chemistry, Graduate School of Engineering, University of Hyogo, 2167 Shosha Himeji, Hyogo 671-2280, Japan. [4] Research Division of Environmental Technology, Osaka Research Institute of Industrial Science and Technology, 1-6-50, Morinomiya, Joto-ku, Osaka 536-8553, Japan. [5] The Electron Microscopy Centre, Tohoku University, 2–1–1 Katahira, Aoba, Sendai 980–8577, Japan. [6] Application & Software Development Department, X-ray Instrument Division, Rigaku Corporation, 3-9-12 Matsubara-cho, Akishima-shi, Tokyo 196-8666, Japan. [7] Graduate School of Engineering, Chiba University, 1-33 Yayoi, Inage, Chiba 263-8522, Japan. [8] Department of Applied Chemistry, The University of Tokyo, 7-3-1 Hongo, Bunkyo-ku, Tokyo 113-8656, Japan. [9] Research Center for Solar Energy Chemistry, Osaka University, 1-3 Machikaneyama, Toyonaka, Osaka 560-8531, Japan. [10] Institute for Materials Chemistry and Engineering, Kyushu University, 744 Motooka, Nishi-ku, Fukuoka 819-0395, Japan. Correspondence and requests for materials should be addressed to H.N. (email: hirotomo.nishihara.b1@tohoku.ac.jp) or to F.T. (email: tanif@ms.ifoc.kyushu-u.ac.jp)

Carbonaceous materials are generally prepared by carbonization of organic substances. During the carbonization process, organic precursors are thermally converted into aggregations of imperfect graphene fragments via intermediates of polycyclic aromatic compounds[1]. Despite their complicated and random structures, carbonaceous materials possess many advantageous properties (electrical conductivity, chemical and thermal stability, light weight). Hence, they are used in a variety of applications including adsorbents, catalysts, supercapacitors, and polymer-electrolyte fuel cells (PEFCs)[2]. When precursors like porphyrins and phthalocyanines with metal/nitrogen (M/N) are employed, their heteroatoms are dispersed in the resulting M/N/C composites[3]. Thus, they show great potential as non-Pt catalysts for oxygen-reduction reactions in PEFCs[4–6]. The thermal conversion process for the production of these carbonaceous materials comprises complex, poorly controlled radical reactions[1]. Hence, the molecular-level control of this process to realize next-generation, high-performance functional carbonaceous materials is challenging. To overcome this, carbonization of molecular-based crystals (metal-organic frameworks[7–16]/molecular organic crystals[17, 18]) with well-designed chemical and supramolecular structures has been employed. This controls the process indirectly via the chemical structures of the precursors while retaining the bulk particle morphology[8, 10] and/or approximate porosity[17], even after carbonization. However, molecular-based crystals convert into intrinsically amorphous carbonaceous frameworks and the precursor structure and molecular features are totally lost during carbonization. Structure-preserving carbonization has only been achieved in mesoscopic organic structures (>5 nm) formed by self-assembly of block-copolymers or surfactant templates[19–21]. Moreover, the direct conversion of organic crystals into ordered carbonaceous frameworks (OCFs) has not been demonstrated.

Herein, we propose the design and supramolecular network structure of a precursor molecule. Our aim was to preserve the precursor structure, while converting a limited part into a carbonaceous framework, to synthesize hybrid materials. These are equipped with precursor structural and chemical features and carbon material properties. The cyclic porphyrin dimer (Ni$_2$-CPD$_{Py}$)[22] met these criteria. Ni$_2$-CPD$_{Py}$ comprises two Ni-porphyrins linked by two butadiyne (diacetylene) moieties through phenyl groups; each porphyrin includes two *meso*-pyridyl groups. The M–N$_4$ (M = metal) unit in the porphyrin ring is thermally stable (~973 K)[23, 24]. Diacetylene is thermally polymerized to poly(diacetylene) to form a rigid crosslinked network, allowing the precursor morphology to be maintained during carbonization. Ni$_2$-CPD$_{Py}$ does not contain volatile fragments[1], thus, a high carbon yield essential to retain the overall framework morphology of the precursor crystal is predicted.

## Results

**Carbonization of Ni$_2$-CPD$_{Py}$.** The molecular structure of Ni$_2$-CPD$_{Py}$ is shown in Fig. 1a. The thermal behaviour of Ni$_2$-CPD$_{Py}$ was investigated and compared to the corresponding free base porphyrin (H$_4$-CPD$_{Py}$)[25]. Figure 1b displays weight changes of Ni$_2$-CPD$_{Py}$ and H$_4$-CPD$_{Py}$ measured by thermogravimetry (TG) in N$_2$ and their differential scanning calorimetry (DSC) curves. The molecules do not exhibit weight loss for temperatures ≤750 K, demonstrating excellent heat-stability and minimal volatility. Thus, Ni$_2$-CPD$_{Py}$ and H$_4$-CPD$_{Py}$ afforded high yields (91% and 77%, respectively, 1073 K). Their carbonization processes were further analysed by temperature-programmed desorption (TPD) with thermogravimetry/photoionization mass spectrometry (TG-PI-MS, Supplementary Fig. 1). Few species (C$_6$H$_6$, C$_5$NH$_5$, C$_7$H$_8$, C$_4$NH$_5$/C$_5$H$_7$, C$_5$H$_8$, and NH$_3$) were desorbed from Ni$_2$-CPD$_{Py}$, while a variety of species were found in H$_4$-CPD$_{Py}$, suggesting that limited decomposition occurs in Ni$_2$-CPD$_{Py}$. The carbonaceous residues afforded after TG measurement were analysed by transmission electron microscopy (TEM, Fig. 1c,d). The product derived from Ni$_2$-CPD$_{Py}$ displayed an ordered structure (periodicity = 14.7 Å). The corresponding electron diffraction pattern clearly differed from that of the graphite (002) plane (periodicity = 3.4 Å). Conversely, the H$_4$-CPD$_{Py}$ residue did not exhibit a highly ordered structure (Fig. 1d). Thus, the porphyrin cation significantly affected the carbonization process and the resulting structure. Ni stabilizes the porphyrin against the thermochemical decomposition, thereby achieving the better yield and retaining the ordered structure.

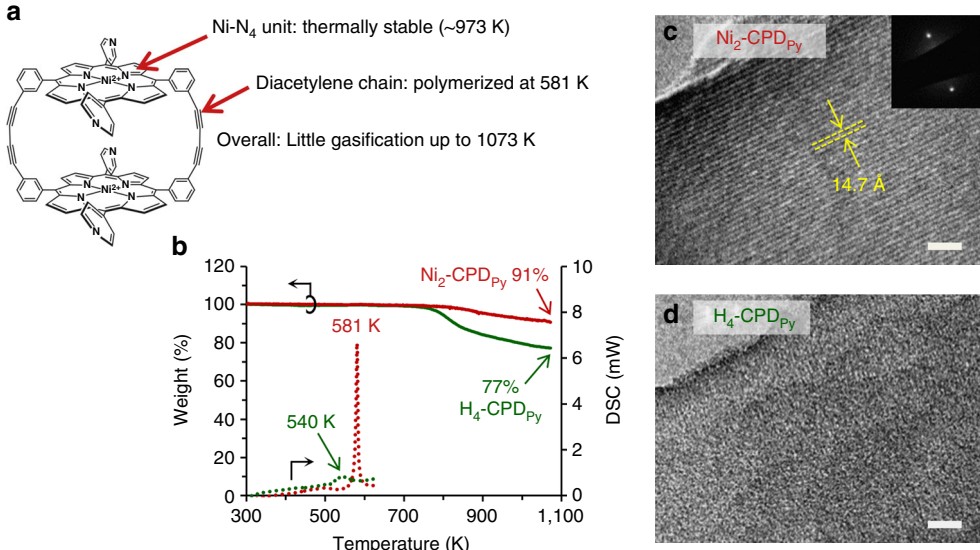

**Fig. 1 Structure of Ni$_2$-CPD$_{Py}$ and its thermal properties up to a temperature of 1073 K. a** Structure of Ni$_2$-CPD$_{Py}$ and its superior properties as a precursor of carbonization. **b** TG (*solid lines*) and DSC curves (*dotted lines*) of Ni$_2$-CPD$_{Py}$ (*red*) and H$_4$-CPD$_{Py}$ (*green*). Yields at 1073 K are described for TG curves, while peak temperatures are shown for DSC curves. **c, d** TEM images of the residues of (**c**) Ni$_2$-CPD$_{Py}$ and (**d**) H$_4$-CPD$_{Py}$ after TG measurements. Scale bars = 10 nm. Inset: a selected-area diffraction pattern for **c**

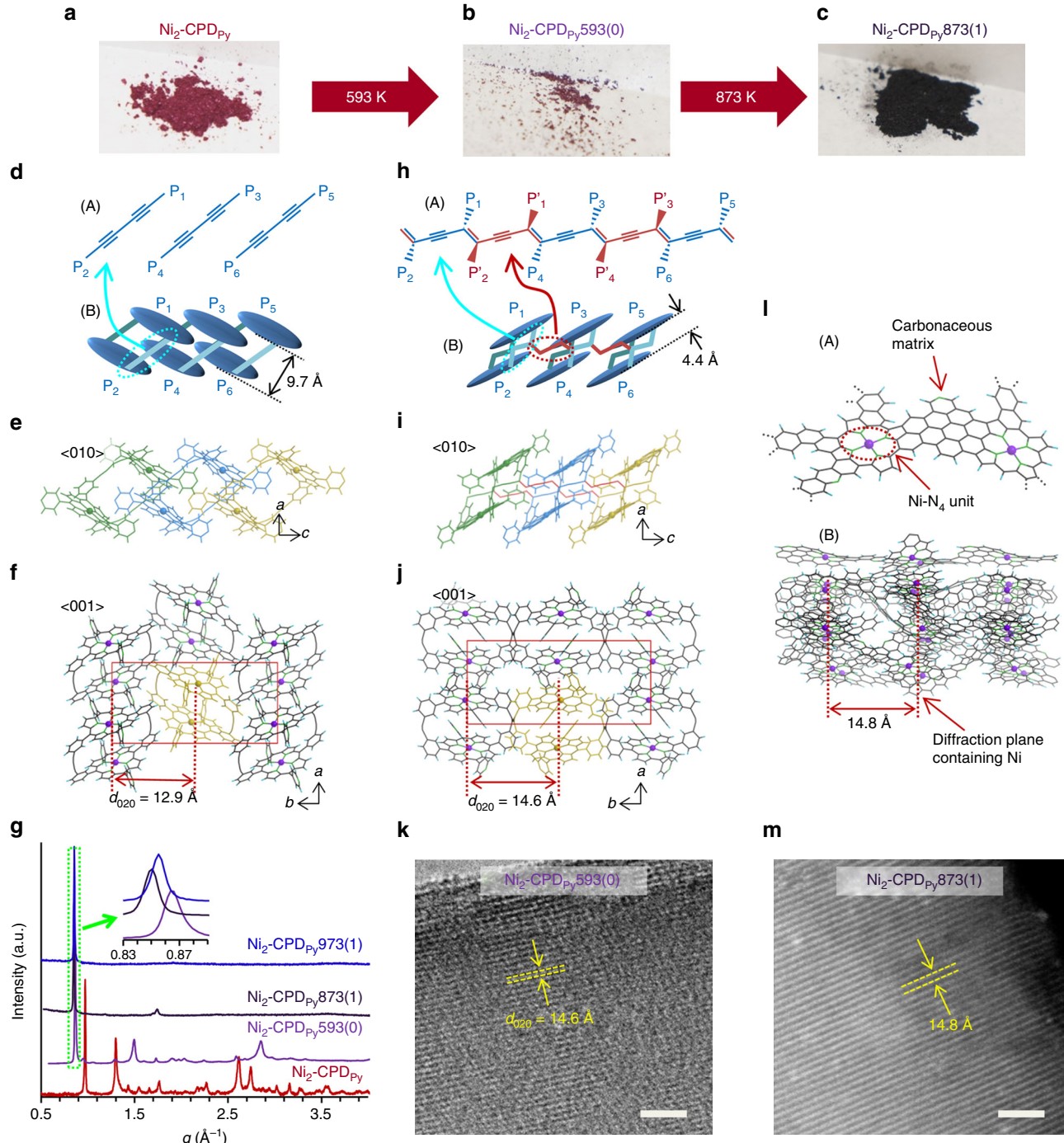

**Fig. 2** Structure evolution of $Ni_2$-$CPD_{Py}$ upon heat-treatment. **a–c** Photographs of (**a**) $Ni_2$-$CPD_{Py}$, (**b**) $Ni_2$-$CPD_{Py}$593(0), and (**c**) $Ni_2$-$CPD_{Py}$873(1). **d** Schematic representation of diacetylene chains (A) and $Ni_2$-$CPD_{Py}$ molecules (B); $P_n$ = porphyrin unit ($n$ = 1–6). **e** Molecular structure of **d**-(B): bottom illustration, <010> direction; each molecule is displayed in a different colour. **f** Larger area view: <001> direction; yellow part corresponds to the yellow moiety in **e**. **g** PXRD patterns of $Ni_2$-$CPD_{Py}$ and heat-treated samples; inset: enlarged intense peaks of heat-treated samples. **h–j** $Ni_2$-$CPD_{Py}$593(0) packing structure. **h** Schematic representation of part corresponding to **d**. In (A), porphyrins locate next to those shown in (B) are indicated with the symbol, $P_n$'. **i** Molecular structure of **h**-(B): <010> direction. Coloured moieties correspond to those in **e**. **j** Larger area view from the <001> direction; yellow part corresponds to the yellow moiety in **i**. Red box in **f** and **j** is unit cell. **k** TEM image of $Ni_2$-$CPD_{Py}$593(0). **l** Expected molecular-level structure of $Ni_2$-$CPD_{Py}$873(1); an enlarged part (A) and a larger region (B) corresponding to **j**. For **f**, **j**, and **l**: C, H, N, O, Ni = *black, light blue, green, red,* and *purple,* respectively. **m** HAADF-STEM image of $Ni_2$-$CPD_{Py}$873(1). Scale bars in **k** and **m** are 10 nm

The $Ni_2$-$CPD_{Py}$ DSC curve (Fig. 1b) exhibits an intense exothermic peak at 581 K (integration = 142 J g$^{-1}$ heat). This corresponds to 102 kJ mol$^{-1}$ per diacetylene amount included in $Ni_2$-$CPD_{Py}$ and this value is ascribed mainly to the diacetylene heat of polymerization (well in agreement with literature values: 80–151 kJ mol$^{-1}$)[26, 27]. The $H_4$-$CPD_{Py}$ DSC curve exhibits a weaker peak at 540 K, affording 41 kJ mol$^{-1}$ per diacetylene. This suggests that the cross-linking in $H_4$-$CPD_{Py}$ is not well developed, resulting in the collapse of the ordered structure (Fig. 1d).

**Crystallographic structural changes upon carbonization.** The structure evolution of $Ni_2$-$CPD_{Py}$ upon heat treatment was analysed to understand the formation mechanism of the fine-ordered structure (Fig. 1c). Figure 2 summarizes the characterization results of $Ni_2$-$CPD_{Py}$ and its heat-treated samples. $Ni_2$-$CPD_{Py}$ almost retains its colour after polymerization [$Ni_2$-$CPD_{Py}$593(0)], indicating the preservation of the porphyrin unit (Fig. 2a,b). After carbonization, the sample turned black (Fig. 2c), confirming the conversion into a carbonaceous substance (graphene sheet formation). We previously reported the single-crystal structure of $Ni_2$-$CPD_{Py}$ accommodating toluene as a guest[22]. Herein, we employed a guest-free crystal (Fig. 2d–f) as a precursor to exclude the effect of toluene and simplify the carbonization process. The crystallographic structure was solved by the direct space method[28–30] and Rietveld refinement[31] of the powder X-ray diffraction (PXRD) pattern (Fig. 2g and Supplementary Fig. 2). The $Ni_2$-$CPD_{Py}$ molecule exhibits a slipped-conformation (Fig. 2d–f). This differed from the overlapped conformation of its single crystal form[22], and the distance between the two porphyrins was determined as 9.7 Å. The $Ni_2$-$CPD_{Py}$ molecules are aligned along the $c$-axis to form columnar arrangement (Fig. 2e), and the columns are integrated to form the structure shown in Fig. 2f (see the detail structure in Supplementary Fig. 3 and Supplementary Movie 1). The diacetylene moieties are located in both sides of the column along the $b$ axis. This ordered arrangement enables solid-phase polymerization to form another crystalline phase (Fig. 2h–j) that was similarly solved (Supplementary Fig. 4). Due to polymerization, the distance between the two porphyrins shortens (4.4 Å) and the $Ni_2$-$CPD_{Py}$ molecules along the $b$ and $c$ axes are cross-linked through a poly(diacetylene) backbone to form a two-dimensional sheet. These sheets are stacked to form a crystal structure (Fig. 2j and Supplementary Fig. 5, Supplementary Movie 2). The polymerization of diacetylene moieties into poly(diacetylene) backbone (Fig. 2d, h) is commonly observed in organic molecules[26, 32]. Solid $^{13}$C NMR confirmed that the diacetylene moieties are almost completely cross-linked to form the poly(diacetylene) form (Supplementary Fig. 6). Thus, $Ni_2$-$CPD_{Py}$593(0) is insoluble in chloroform (good solvent for $Ni_2$-$CPD_{Py}$) and $^1$H NMR and matrix-assisted laser desorption ionization time-of-flight mass spectrometry (MALDI-TOF-MS) did not detect any remaining monomer/oligomers. The transformation from monomer crystal into a crystalline covalent-organic framework is ascribed to the close proximity of the diacetylene moieties in $Ni_2$-$CPD_{Py}$ (Fig. 2d–f), allowing solid-phase polymerization to proceed. On the other hand, $H_4$-$CPD_{Py}$ is not a highly crystalline solid, and its packing structure cannot be solved from the PXRD pattern in its guest-free form (Supplementary Fig. 7). $H_4$-$CPD_{Py}$ has broad PXRD peaks, and it means that the solid contains irregular packing structures and distributed distances between diacetylene moieties, causing imperfect polymerization. Thus, the original packing structure collapses during pyrolysis (Supplementary Fig. 7).

Upon polymerization, the $d$-spacing (12.9 Å) of the original $Ni_2$-$CPD_{Py}$ (020) plane increases to 14.6 Å (Fig. 2g) and its periodicity is clearly observed in the TEM image (Fig. 2k). The structure regularity of the (020) plane is well retained, even after heat treatment (873 K). This is confirmed by the sharp peak (inset of Fig. 2g; $d$-spacing = 14.8 Å) that exhibits a $d$-spacing that is slightly greater than that of its precursor polymer (14.6 Å). Generally, the carbonization of organic substances results in matrix shrinkage[19, 20] because graphene and its stacked graphitic structure have atomically denser frameworks. Thus, the increase in $d$-spacing indicates the formation of low-density framework structures. The resulting $Ni_2$-$CPD_{Py}$873(1) was further analysed by high angle annular dark-field scanning transmission electron

microscopy (HAADF-STEM; Fig. 2m: white-coloured area = Ni atoms). Ni rarely forms nanoparticles or aggregates often generated in the carbonaceous residues of metal porphyrins[33, 34] (an example is shown later in the carbonization of 5,10,15,20-tetraphenyl-21H,23H-porphine nickel(II) [Ni-TPP]). Ni and NiO formation was not detected, even by synchrotron PXRD analysis (Supplementary Fig. 8). The image in Fig. 2m well agrees with the TEM image (Supplementary Fig. 9). Notably, the ordered structure spreads to several hundred nanometers (Supplementary Fig. 9). The results of TEM, HAADF-STEM, and synchrotron PXRD reveal that Ni is not aggregated as Ni metal or NiO, and exists along the structure regularity derived from the (020) plane of its precursor. As shown later, Ni retains its original coordination structure (Ni-$N_4$) in the carbonaceous framework. Moreover, synchrotron PXRD (Supplementary Fig. 8) and fast Fourier transform TEM images (Supplementary Fig. 10) prove the presence of several diffraction planes other than the (020) plane and high-order planes in $Ni_2$-$CPD_{Py}$873(1). This indicates its well-ordered structure, similar to organic-based frameworks. Fig. 2l displays a possible $Ni_2$-$CPD_{Py}$873(1) framework built on experimental evidence (see Supplementary Methods about the details of model construction). Porphyrin moieties are linked by a carbonaceous matrix keeping the Ni–$N_4$ structure and their approximate original positions [$Ni_2$-$CPD_{Py}$593(0)]. Thus, Ni atoms form the diffraction plane observed by PXRD and TEM/STEM analysis. As represented by this model (3D view is provided in Supplementary Movie 3), $Ni_2$–$CPD_{Py}$873(1) exhibits an intermediate framework between an organic substance and carbon, making this material distinguishable from any other substances. Moreover, the sample carbonized at 973 K still retains structural regularity with very small shrinkage (Fig. 2g, $d$-spacing = 14.7 Å) confirming the excellent heat stability of the ordered structure.

**Chemical structure transition upon carbonization.** The change in elemental composition associated with carbonization is summarized in Table 1. The amount of hydrogen decreases with an increase in temperature suggesting growth of the graphene sheets. Generally, carbons formed at 873 K are still defective and include dangling bonds, which are oxidized upon exposure to air. Hence, $Ni_2$-$CPD_{Py}$873(1) contains a small amount of oxygen. At a higher carbonization temperature (973 K), the oxygen amount decreases, indicating a decrease in dangling bonds. This is attributed to further development of the graphene sheets. Notably, the N and Ni contents are well retained up to 973 K. Thus, a hybrid material containing a large fraction of heteroatoms can be produced.

The $Ni_2$-$CPD_{Py}$ chemical structure transition was analysed by Raman spectroscopy (Fig. 3a). Some of the major peaks in the Raman spectrum of $Ni_2$-$CPD_{Py}$ can be ascribed as follows: breathing, 1010 $cm^{-1}$; $\delta$(C–H), 1230 and 1355 $cm^{-1}$; $\nu$(C=C), 1500 and 1590 $cm^{-1}$; $\nu$(C≡C), 2130 $cm^{-1}$; and $\nu$(C–H), 3110 $cm^{-1}$, from the results of simulation using the Gaussian 09 software[35], (Supplementary Methods, Supplementary Figs. 11

**Table 1 Elemental compositions of the samples**

| Sample | Elemental composition (wt%) | | | | |
| --- | --- | --- | --- | --- | --- |
| | C | H | N | Ni | O |
| $Ni_2$-$CPD_{Py}$[a] | 76.8 | 3.4 | 11.7 | 8.2 | 0 |
| $Ni_2$-$CPD_{Py}$873(1) | 75.7 | 2.0 | 10.5 | 9.1 | 2.7 |
| $Ni_2$-$CPD_{Py}$973(1) | 78.1 | 1.1 | 10.2 | 9.4 | 1.2 |

[a]For $Ni_2$-$CPD_{Py}$, the composition is calculated from its molecular formula.

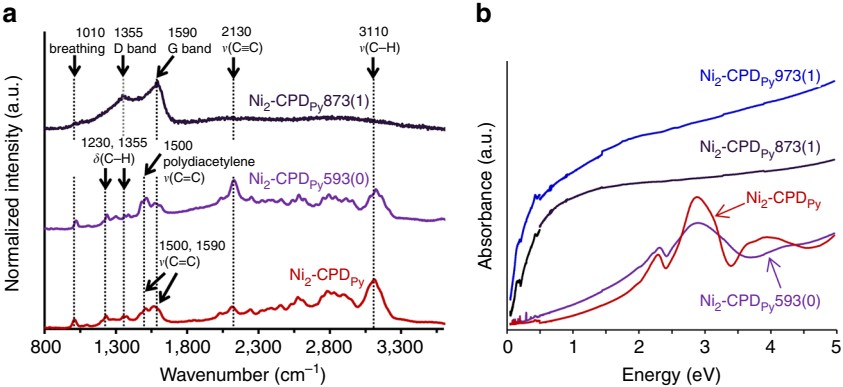

**Fig. 3** Raman and UV-vis absorption spectra of $Ni_2$-$CPD_{Py}$ and heat-treated samples. **a** Raman spectra: peak assignments are based on theoretical calculations (Supplementary Fig. 12) and references[36, 39]. **b** Absorption spectra measured by FT-IR (<0.4959 eV) and UV-vis-NIR spectrometry (>0.4959 eV)

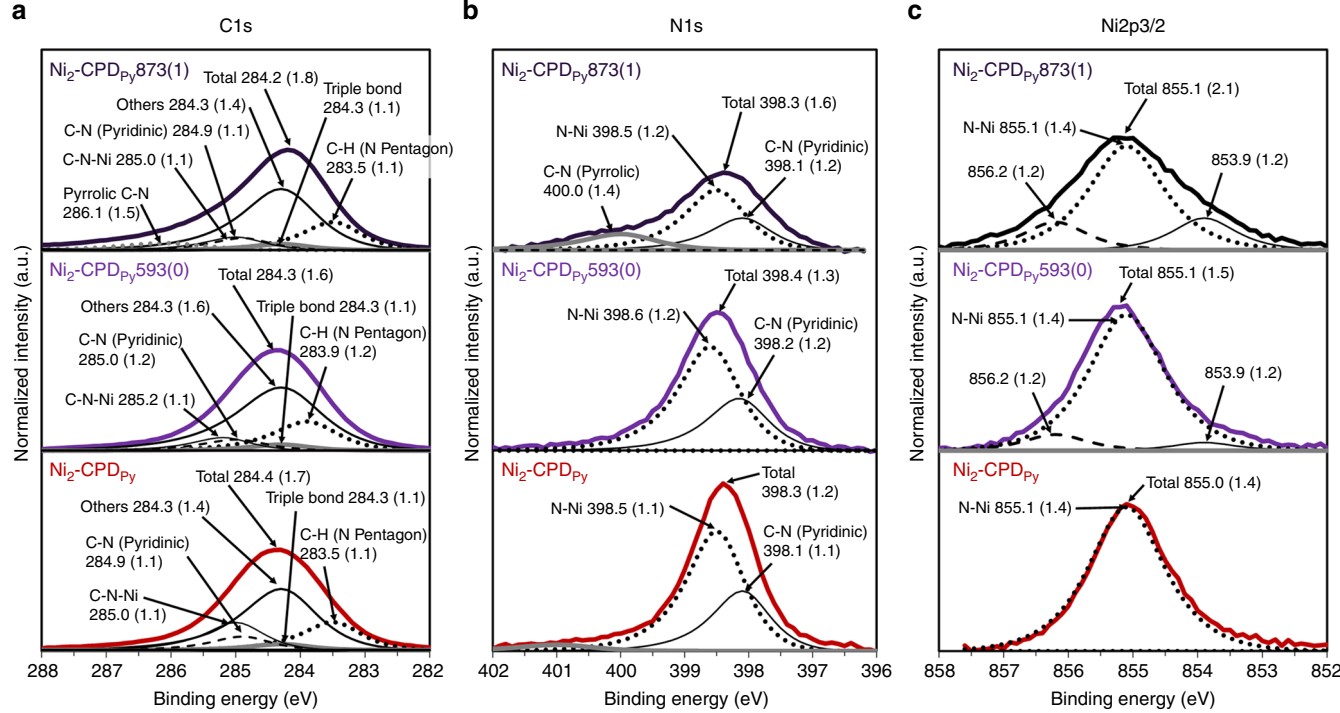

**Fig. 4** X-ray photoelectron spectroscopy results of $Ni_2$-$CPD_{Py}$ and heat-treated samples. **a** C1s, **b** N1s, **c** Ni2p$_{3/2}$. Numbers before parenthesis indicate binding energy in eV. Numbers inside the parenthesis indicate FWHM of spectra. In **a** and **b**, experimentally obtained peaks are deconvoluted into several peaks determined by theoretical calculations (Supplementary Fig. 16). In **c**, FWHM of the N–Ni peak is determined as 1.4 eV from the result of $Ni_2$-$CPD_{Py}$, and the broadened parts in the heat-treated samples were deconvoluted into three peaks including two additional peaks that have lower (853.9 eV) and higher (856.2 eV) binding energies than that of N–Ni

and 12). Most peaks are retained in $Ni_2$-$CPD_{Py}$593(0); this is in agreement with the structure change shown in Fig. 2. A peak at 1500 cm$^{-1}$ becomes intense in the polymer, and this reflects the formation of a poly(diacetylene) backbone (Supplementary Fig. 12). In $Ni_2$-$CPD_{Py}$873(1), most peaks disappear, and only broad D- and G-bands[36] originating from intervalley scattering[37–39] appear, indicating that the well-defined chemical structure of the precursor polymer framework is lost and the resulting OCF consists of defective graphene sheets (Fig. 2l), like zeolite-templated carbons (ZTCs) and ordered mesoporous carbons[2]. $Ni_2$-$CPD_{Py}$ absorption spectrum (Fig. 3b) displays Q and Soret bands of the $Ni^{2+}$ porphyrin unit (2.31 and 2.89 eV, respectively). Almost no energy shift is observed in $Ni_2$-$CPD_{Py}$593(0), strongly suggesting that the electric structures

of these units are retained in $Ni_2$-$CPD_{Py}$593(0). This agrees with the structure in Fig. 2h–j. Conversely, $Ni_2$-$CPD_{py}$873(1) and $Ni_2$-$CPD_{py}$973(1) display a broad absorption band from the mid-IR to UV region assigned to the interband transitions of the graphene sheets. Since the absorption of this transition is too strong, the absorption bands of the $Ni^{2+}$ porphyrin unit are veiled and their presence cannot be confirmed in Fig. 3b. As shown later, the chemical states of Ni in the carbonized samples were analyzed also by X-ray absorption fine structure (XAFS) measurements of the Ni–K edge.

The formation of graphene sheets renders the OCFs electrically conductive. At room temperature, $Ni_2$-$CPD_{py}$ and $Ni_2$-$CPD_{py}$593 (0) were highly insulating (resistivity $\rho > 1$ TΩ cm). Meanwhile, in $Ni_2$-$CPD_{py}$873(1) and $Ni_2$-$CPD_{py}$973(1), a significant decrease

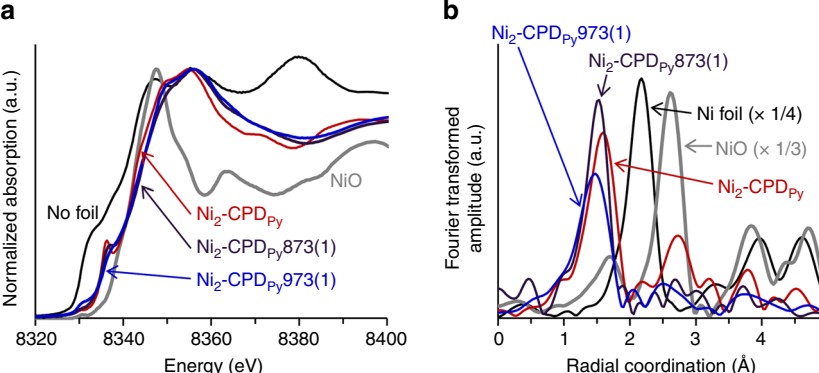

**Fig. 5** X-ray absorption fine structure results of $Ni_2$-$CPD_{Py}$ and its heat-treated samples. **a** XANES spectra. **b** Pseudo-radial structural functions calculated from EXAFS patterns. The data of Ni foil and NiO are also shown for comparison

in $\rho$ (33.8 and 18.6 $\Omega$ cm, respectively at room temperature) was observed. Thus, unlike conventional organic-based frameworks, OCFs are equipped with electric conductivity. As the temperature decreased, both samples exhibited an increase in resistivity, indicating that both are semiconductors (Supplementary Fig. 13). The activation energies were estimated as 0.050–0.091 eV for $Ni_2$-$CPD_{py}873(1)$ and 0.015–0.022 eV for $Ni_2$-$CPD_{py}973(1)$. The microstructures of OCFs were further studied by X-band electron paramagnetic resonance (EPR) analysis (Supplementary Fig. 14). $Ni_2$-$CPD_{Py}873(1)$ at 5.0 K displayed an absorption line ($g \sim 2$) with a sharp line-width (2.8 mT). These results indicate the formation of organic radicals derived from dangling bonds generated during the carbonization process. Conversely, $Ni_2$-$CPD_{Py}973(1)$ exhibited broad absorptions and no signals attributed to radicals were observed. The broad peak is assigned to coupling with anisotropic spins in the six-coordinated paramagnetic $Ni^{2+}$ ions generated during heat treatment. The structural change of $Ni_2$-$CPD_{Py}$ was further analysed by X-ray photoelectron spectroscopy (XPS, Fig. 4). No significant changes were observed in the C1s, N1s, and $Ni2p_{3/2}$ XPS spectra during the transition from $Ni_2$-$CPD_{Py}$ to $Ni_2$-$CPD_{Py}593(0)$. This is explained by the change in structure shown in Fig. 2. The C1s spectra of $Ni_2$-$CPD_{Py}$ and $Ni_2$-$CPD_{Py}593(0)$ are well in agreement with results from theoretical calculations (Supplementary Figs. 15 and 16)[35, 40–44] as well as the positions and full width at half maximum (FWHM) of the deconvolution peaks. Moreover, the C1s, N1s, and $Ni2p_{3/2}$ spectra of $Ni_2$-$CPD_{Py}873(1)$ indicate that the chemical environment of these atoms differ only slightly from those in $Ni_2$-$CPD_{Py}593(0)$, despite the disappearance of the well-defined phenyl groups, C≡C, and C–H bonds (Fig. 3a). In $Ni_2$-$CPD_{Py}873(1)$, the N1s spectrum slightly broadens (change in FWHM) because of the appearance of a small C–N (pyrrolic) component at 400.0 eV. These moieties may be formed by thermal conversion of the pyridyl groups into the carbonaceous framework containing the pyrrolic structure, or by cleavage of porphyrin rings followed by hydrogen addition to the free pyrrolic N. The $Ni2p_{3/2}$ spectrum of $Ni_2$-$CPD_{Py}$ broadens after heat treatment (two peak components at 856.2 and 853.9 eV). These components are ascribed to the oxidized Ni species, carbides, or metallic Ni[45].

**Chemical environment of Ni in the carbon matrix.** The XAFS of the Ni–K edge was also analysed by synchrotron X-ray absorption spectroscopy [X-ray absorption near edge structure (XANES) spectra in Fig. 5a]. The Ni–K edge energy (~8333 eV) of $Ni_2$-$CPD_{Py}$ lies between those of Ni foil (8329 eV) and NiO (8336 eV). This reflects the intermediate oxidation state of Ni in $Ni_2$-$CPD_{Py}$. The $Ni_2$-$CPD_{Py}$ XANES spectrum exhibits a characteristic peak at

8336 eV, corresponding to the 1 s to $4p_z$ transition[46]. This is typical of a planar porphyrin[47] and phthalocyanine[46] where the Ni coordinates with four nitrogen atoms in the Ni–$N_4$ unit. Notably, the $Ni_2$-$CPD_{Py}873(1)$ XANES spectrum is almost unchanged from that of $Ni_2$-$CPD_{Py}$, indicating the retention of the Ni–$N_4$ unit even after carbonization. In $Ni_2$-$CPD_{Py}973(1)$, a shoulder appears at 8329 eV, suggesting the formation of a small amount of metallic Ni due to the partial decomposition of the porphyrin moieties. However, the overall spectrum is still well retained.

The pseudo-radial structure functions were next calculated using the Ni–K edge extended X-ray absorption fine structure (EXAFS) spectra for $Ni_2$-$CPD_{Py}$ and its carbonized derivatives, Ni foil, and NiO (Fig. 5b). $Ni_2$-$CPD_{Py}$ exhibits an intense peak at 1.56 Å, corresponding to the four N atoms coordinated to the Ni atom. The carbonized samples display very similar patterns to that of $Ni_2$-$CPD_{Py}$, confirming the retention of the Ni–$N_4$ unit after carbonization. The precise distance between Ni and N and its coordination number were calculated with FEFF8.2 (Supplementary Table 1). $Ni_2$-$CPD_{Py}873(1)$ and $Ni_2$-$CPD_{Py}973(1)$ retained the coordination numbers 3.8 and 3.4, respectively. This is close to the initial number (4).

The chemical environment of the $Ni^{2+}$ ions was studied also by magnetic susceptibility analysis (Supplementary Fig. 17). Since $Ni^{2+}$ ions in the square planar coordination geometry are diamagnetic, $Ni_2$-$CPD_{Py}$ and $Ni_2$-$CPD_{Py}593(0)$ seldom respond to an external magnetic field. Their minute responses are ascribed to minor impurities. $Ni_2$-$CPD_{Py}873(1)$ retains a weak magnetization, indicating that the square planar coordination is well retained in the sample; this is in agreement with XAFS data. These samples almost obey the Curie-Weiss law in the range of 5.0–300 K, with small Curie constants ($C = 6.3 \times 10^{-6}$, $15 \times 10^{-6}$ and $87 \times 10^{-6}$ cm$^3$ g$^{-1}$ K, respectively). Conversely, $Ni_2$-$CPD_{Py}973(1)$ data deviate from this law. This is attributed to the temperature-independent paramagnetic term, $\chi_p$, corresponding to Pauli's paramagnetism of the graphene sheets and Ni metal cluster and agrees with results from XANES (Fig. 5a; curve fitting: $\chi_p = 13.7 \times 10^{-6}$ cm$^3$ g$^{-1}$ and $C = 202 \times 10^{\times 6}$ cm$^3$ g$^{-1}$ K). The increase in $C$ is attributed to the generation of six-coordinated paramagnetic $Ni^{2+}$ ions formed by heat treatment. If all the $Ni^{2+}$ ions are converted into six-coordinated species, then $C = 1391 \times 10^{-6}$ cm$^3$ g$^{-1}$ K ($S = 1$, $g = 2.0$). Since the value of $Ni_2$-$CPD_{Py}973(1)$ is 14.5% of this assumption, 85.5% of $Ni^{2+}$ ions are expected to retain the square-planar coordination geometry, even after heat treatment at 973 K.

**Electrochemical catalysis.** Compared to carbon materials, organic-based frameworks have a great advantage of chemical

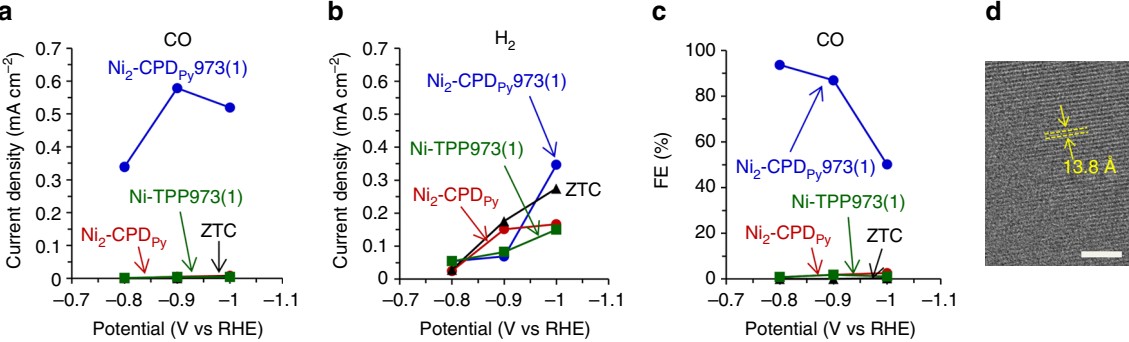

**Fig. 6** Examination of selective $CO_2$ electro-reduction into CO. **a**, **b** Partial current densities used for (**a**) CO and (**b**) $H_2$ generation on the samples in $CO_2$-saturated 0.1 M $KHCO_3$. **c** FE for CO generation. **d** TEM image of a reference ordered microporous carbon, ZTC

designability. In the latter materials, specific molecular blocks can be integrated three-dimensionally with structure order, by which a variety of unique functions can be achieved. However, they are intrinsically not electrically conductive unlike carbon, and are necessarily deposited as thin films on good conductors for electrochemical applications[48–50]. The proposed OCFs are expected to provide new electrocatalyst designs in which active sites are embedded in electrically conductive frameworks with structure regularity. To prove this concept, we have examined the electrocatalytic activity of the porphyrin center ($Ni–N_4$ unit) retained in $Ni_2$-$CPD_{Py}$973(1). Unlike the cases of Fe and Co, Ni-based complexes are generally poor in electrocatalysis for oxygen-reduction reaction. However, specific Ni cyclam complexes[51–54] or Ni-N-modified graphene[55], which have $Ni–N_4$ sites, have been reported to show unique electrocatalysis for the $CO_2$ reduction into CO without significant $H_2$ evolution and with a high Faradaic efficiency (FE) of ca. 90%. Additionally, this selective $CO_2$ reduction has rarely been reported in carbonaceous Ni/N/C materials made from Ni-based complexes. Figure 6 summarizes the comparison of selective $CO_2$ reduction activities of $Ni_2$-$CPD_{Py}$973(1) and three reference materials, characterized by the method reported elsewhere[55]. $Ni_2$-$CPD_{Py}$973(1) shows apparent $CO_2$ reduction catalysis into CO (Fig. 6a) even without any conductive additives, while the activity for $H_2$ evolution is quite low below −0.9 V vs. RHE (reversible hydrogen electrode) (Fig. 6b). Thus, $Ni_2$-$CPD_{Py}$973(1) achieves high FE of 94 and 87% at −0.8 and −0.9 V, respectively (Fig. 6c). To further investigate the uniqueness of the OCF catalysis, the same measurement was applied to three reference materials: ZTC, $Ni_2$-$CPD_{Py}$, and Ni-TPP carbonized at 973 K for 1 h [Ni-TPP973(1)]. ZTC is an existing ordered microporous carbon prepared by using zeolite as a hard template[2], and has the structure periodicity of 13.8 Å (Fig. 6d), which is close to that of $Ni_2$-$CPD_{Py}$973(1). ZTC has an electrically conductive framework which comprises mainly of $sp^2$ carbons[56], but it does not possess catalysis sites including metal species. ZTC shows no catalysis in Fig. 6a, and this result clearly indicates that the ordered carbonaceous framework itself can never reduce $CO_2$. $Ni_2$-$CPD_{Py}$ also shows no catalysis in Fig. 6a despite the presence of the $Ni–N_4$ unit, because the organic crystal of $Ni_2$-$CPD_{Py}$ is not electrically conductive. Next, we discuss the active site. In the cases of Fe/N/C and Co/N/C electrocatalysts for oxygen reduction reaction, catalysis sites often exist as disordered forms consisting of metal species, N, and C. Hence, it is necessary to examine the catalytic activity of disordered Ni/N/C structure towards the selective $CO_2$ reduction into CO. For this purpose, Ni-TPP973(1) was prepared as a representative Ni/N/C material by the same carbonization procedure as that for $Ni_2$-$CPD_{Py}$973(1), from a common Ni-based porphyrin Ni-TPP. During the heat treatment, Ni-TPP is decomposed involving cleavage of Ni–N bonds, and it

turns into the mixture of a disordered Ni/N/C framework and Ni metal aggregation (Supplementary Fig. 18). The resulting Ni-TPP973(1) shows no catalysis in Fig. 6a, indicating that the disordered Ni/N/C structure is not active, and the $Ni–N_4$ unit is the active site in $Ni_2$-$CPD_{Py}$973(1). Figure 6 thus proves that the selective $CO_2$ reduction catalysis of $Ni_2$-$CPD_{Py}$973(1) can be achieved by the intermediate structure of organic-based frameworks and carbon materials, in which molecular catalysis sites ($Ni–N_4$) are embedded in the conductive framework. As mentioned above, conventional M/N/C catalysts have disordered structures and this has hampered the basic understanding between the structure and catalysis. With its certainly determined catalysis sites, $Ni_2$-$CPD_{Py}$973(1) can be a good platform to investigate the fundamentals of carbonaceous electrocatalysts.

The advantage of OCFs compared to conventional carbon materials is chemistry-based better controllability. Though the present OCFs are poorly porous (Supplementary Table 2), it is possibly improved by introducing volatile groups at designed sites of the starting molecules. Replacing Ni with other metals such as Fe, Co, Cu, Pt, and Pd can also widen the versatility of OCFs. Moreover, the exterior shape of the crystal can be also controlled based on the existing methods, for example a reprecipitation method[57–60] (Supplementary Fig. 19) to achieve additional function[59]. By their chemical designability, OCFs are expected to be further developed hereafter.

## Discussion

In summary, the direct conversion of organic crystal into OCFs was demonstrated. The successive thermal conversion process actually comprises two steps. A molecular crystal of $Ni_2$-$CPD_{Py}$ is first thermally converted into a crystalline covalent-organic framework, and is further converted into OCFs that inherit the periodic structure and the $Ni–N_4$ unit in the precursor organic crystal. The successful conversion is due to the following properties of $Ni_2$-$CPD_{Py}$. First, absence of volatile moieties like paraffin structures, oxygen, halogens, and sulphur. Second, presence of well-arrayed diacetylene moieties that can be thermally crosslinked to form a heat-stable polymer. Third, the presence of thermally stable (~ 973 K) organic moiety (metal porphyrin unit). On the basis of this strategy, a variety of OCFs could be synthesized, probably also from organic molecules other than $Ni_2$-$CPD_{Py}$. This new pathway allows the preparation of OCFs with molecularly controlled chemical structures that can be considered fusion materials of organic-based frameworks and carbon materials.

## Methods
**Materials**. $Ni_2$-$CPD_{Py}$ and $H_4$-$CPD_{Py}$ were synthesized according to literature[22, 25]. Heat treatment of $Ni_2$-$CPD_{Py}$ and $H_4$-$CPD_{Py}$ was performed at a heating rate of 5 K $min^{-1}$ ramped at a designed temperature (593, 873, or 973 K, $N_2$ flow) by using a

tubular furnace. In the case of 593 K, heating was stopped immediately when the temperature reached 593 K. The samples thus obtained are $Ni_2$-$CPD_{Py}593(0)$ and $H_4$-$CPD_{Py}593(0)$. The sample name is expressed as follows: M-$CPD_{Py}X(Y)$ for M=$Ni_2$ or $H_4$, X is the treated temperature (K), and Y is the period of the treatment (h). In the case of 873 and 973 K, the temperature was maintained at the target temperatures for 1 h. As a reference, 5, 10, 15, 20-tetraphenyl-21H,23H-porphine nickel(II) (≥95%, Sigma-Aldrich) was carbonized at 973 K for 1 h by the same manner as that for $Ni_2$-$CPD_{Py}973(1)$. The sample thus obtained is Ni-TPP973(1). Zeolite-templated carbon was synthesized by the method reported elsewhere[61].

**Characterization**. Porphyrin TG curves were measured by a Shimazu TGA-51 thermogravimetric analyser ($N_2$ flow, ≤1073 K, $H_4$-$CPD_{Py}$/$Ni_2$-$CPD_{Py}$). TPD patterns were measured using a Rigaku ThermoMass Photo spectrometer (10 K min$^{-1}$, ≤1073 K, He flow). The afforded residue was observed by TEM (JEM-2010/JEM-2200FS, JEOL). DSC curves were recorded on a Mettler DSC1 STARe (≤623 K, 10 K min$^{-1}$, $N_2$ flow). To confirm the presence of the monomer/oligomers in $Ni_2$-$CPD_{Py}593(0)$, the sample was dispersed in chloroform and analysed by $^1$H NMR (Bruker Avance III 400) and MALDI-TOF-MS (Bruker Autoflex Speed). $^{13}$C CP-MAS NMR spectra were measured on a JEOL JNM-ECA800 (800 MHz) spectrometer. The sample was packed into a 2.5-mm zirconia rotor, and the measurement was carried out using two-pulse phase-modulated decoupling (30 kHz, 1 s recycle delay, 3 ms contact time, π/2 pulse width = 2.43 µs at 69.5 W, and 2048 scans). The $CH_2$ peak of the external adamatane standard was 29.5 ppm. Spectra were processed with Delta NMR Software (v 5.0) using conventional techniques (100 Hz line broadening window function). The position of Ni in $Ni_2$-$CPD_{Py}873(1)$ was analysed by the HAADF-STEM technique (Titan$^3$ G2 60–300 Double Cs-Corrector, FEI Company; 300 V). Elemental analysis of the carbonized samples was carried out with a Yanaco JM10 analyser. The sample underwent combustion (flow = 20%-$O_2$ + 80%-He); gasses generated were converted into $CO_2$, $H_2O$, and $N_2$ to determine the respective C, H, and N amounts; Ni remained as oxidized ash. The Ni content was determined by assuming its composition as NiO; the amount of O was calculated by subtracting the amount of C, H, N, and Ni from the initial sample weight. Raman spectra were measured with a Jasco NRS-3100 (532.2 nm line). Absorption spectra were measured from samples moulded into KBr pellets (Nicolet 6700 FT-IR spectrometer, 0.04949–0.4959 eV/Perkin-Elmer Lambda750A UV-vis-NIR spectrometer, 0.4959–4.959 eV). The temperature-dependent electric resistance was measured with a two-electrode method for high resistance samples [$Ni_2$-$CPD_{Py}$, $Ni_2$-$CPD_{Py}593(0)$, $Ni_2$-$CPD_{Py}873(1)$; pellet diameter = 3 mm] and a four-probe method for low resistance $Ni_2$-$CPD_{Py}973(1)$ (rod-shape: 3.0 × 0.68 × 0.37 mm). $Ni_2$-$CPD_{Py}873(1)$ and $Ni_2$-$CPD_{Py}973(1)$ were mixed with 5 wt% binder polymer (PTFE). The sample was placed in a Sumitomo SRDK-101D cryogenic refrigerating system. Electric contacts were prepared using Tokuriki #8560 gold paste and 25 µm gold wires. In the two-probe method, force-voltage current measurements were performed using a Keithley 6517 A electrometer. In the four-probe method, a constant current (0–2 µA) was applied (Advantest R6161). The voltage was measured by a Hewlett-Packard 3458 A digital multimeter. XPS spectra were measured with a JEOL JPS-9200. To avoid charge build-up, a solution of $Ni_2$-$CPD_{Py}$ in chloroform was spin-coated on an Al substrate (purity = 99.999, Al-Kα radiation, spot size = 3 mm). The substrate was heat treated (≤593 K) to prepare $Ni_2$-$CPD_{Py}593(0)$ and its XPS spectra were recorded. Subsequently, the substrate was heat treated (873 K, 1 h) to prepare $Ni_2$-$CPD_{Py}873(1)$ and its XPS spectra were recorded. Ni–K edge XAFS measurements, before and after carbonization of $Ni_2$-$CPD_{Py}$, were performed in transmission mode (in air, room temperature, synchrotron radiation BL14B2 beam line, SPring-8). The recorded spectra were normalized and fitted by REX2000 (Rigaku). The precise Ni to N distance and coordination number were calculated from the EXAFS results by using FEFF8.2. Magnetic susceptibility data were collected in the temperature range of 5.0–300 K in an applied field of 10 kG using a Quantum Design MPMS2 SQUID magnetometer. X-band EPR data were recorded on a JEOL JES-FA100 spectrometer equipped with an Oxford ESR900 continuous-flow liquid He cryostat. $N_2$ and $CO_2$ adsorption isotherms were measured at 77 K and 298 K, respectively (Micro-tracBEL Corp. BELMAX). In the $N_2$ adsorption isotherm, the specific surface area was calculated by the Brunauer–Emmett–Teller (BET) method in the pressure range of $P/P_0$ = 0.05–0.35, and the total pore volume ($V_{N2}$) was calculated at $P/P_0$ = 0.96. In the $CO_2$ adsorption isotherm, the pore volume ($V_{CO2}$) was calculated by the Dubinin–Radushkevich equation.

**Data availability**. Crystallographic data (CIF files) for $Ni_2$-$CPD_{Py}$ and $Ni_2$-$CPD_{Py}593(0)$ have been deposited with the Cambridge Crystallographic Data Centre as supplementary publications. CCDC 1552441 ($Ni_2$-$CPD_{Py}$) and CCDC 1552442 ($Ni_2$-$CPD_{Py}593(0)$) contain the supplementary crystallographic data. These data can be obtained free of charge from the Cambridge Crystallographic Data Centre via www.ccdc.cam.ac.uk/data_request/cif. PXRD analysis, construction of a model structure of Ni2-CPDPy873(1), computational simulations, and CO2 reduction measurements are provided in Supplementary Methods. All other data supporting the findings of this study are available within the article and its Supplementary Information.

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

## Acknowledgements

This work was supported by PRESTO, JST; JSPS KAKENHI Grant Number 15KK0196; the Dynamic Alliance for Open Innovation Bridging Human, Environment and Materials; and the Network Joint Research Centre for Materials and Devices. The PXRD and XAFS measurements were performed in SPring-8 (Proposals no. 2015A1956, 2015A1666, and 2016A1750). We thank Tohoku University Molecule and Material Synthesis Platform in Nanotechnology Platform Project, for $^{13}C$ CP-MAS NMR analysis operated by Mr. Shinichiro Yoshida. The authors acknowledge Prof. T. Kyotani for his advice, Dr A. Castro Muniz for his experimental support, Prof. M. Kakihana for performing Raman spectroscopy, Rigaku Co. for their kind support for TPD measurements with TG-PI-MS. The authors express the deepest gratitude to Professor K. Kaneko for his valuable advice.

## Author contributions

H.T.N. designed the project, summarized all the data provided by co-authors, and wrote the manuscript. F.T. synthesized Ni₂-CPD$_{Py}$ and H₄-CPD$_{Py}$ and supervised the outline of this work from the viewpoint of organic chemistry. H.F.N. developed the synthetic methods of Ni₂-CPD$_{Py}$ and H₄-CPD$_{Py}$, and further improved the methods to provide sufficient amount of these substances. T.T.H. carried out TG of organic crystals, and found the retention of the ordered structure by TEM. Then, he characterized the materials with Raman spectroscopy and XPS. K.M. carried out elemental analysis of the materials. T.K. taught H.T.N. and T.T.H. the synthesis of Ni₂-CPD$_{Py}$ and H₄-CPD$_{Py}$, and also contributed to the preparation of their solvent-free crystals. M. O. carried out DSC measurements and gas adsorption measurements. T.K.H. analysed OCFs by high-resolution TEM together with fast Fourier transform, and H.J. supervised the experiment and data analysis. T.A. suggested and designed the analysis of materials by electrical conductivity, EPR, Magnetic susceptibility, and UV-vis absorption spectroscopy, and N.H. carried out these measurements and data analyses. H.T.K. suggested the control of crystal morphology by using the reprecipitation method, and Y.K. carried out the experiment. J.M. carried out XAFS and electrochemical measurements, as well as the data analyses. Y.M. contributed the structural analysis of the carbonized samples. Y.H. performed STEM-HAADF observations of OCFs and analysed the data. H.S.K. revealed the crystal structures of Ni₂-CPD$_{Py}$ and Ni₂-CPD$_{Py}$593(0) from their PXRD patterns. Y.Y. carried out the simulation of Raman spectra and XRS spectra of the samples and analysed the experimental data according to the calculation results. K.K. designed the electrochemical $CO_2$ reduction reaction, and S.Y. performed the measurements and analyses.

## Additional information

**Competing interests:** The authors declare no competing financial interests.

