## [Peer Review File · Nature Communications]

Reviewers' comments:

Reviewer #1 (Remarks to the Author):

The authors emphasize that the resulting ordered carbonaceous frameworks exhibit an intermediate structure, between organic-based frameworks and carbon materials, with advantages pertaining to both structures. We STRONGLY request the authors to show any unique properties derived from such an intermediate structure. The authors try to characterize the obtained materials by lots of tools, I appreciate. Despite the overall high quality of the characterization, I have the negative impression that the submitted manuscript does not fulfill the impact criterium required for a publication in Nature Communications, however.

Reviewer #2 (Remarks to the Author):

This is a very interesting paper for preparing crystalline carbonaceous materials via pyrolysis. It can be published after consideration of the following issues:

- (1) What's the role played by Ni ions? Can other metal ions (e.g., Fe) work?
- (2) What's the surface area of the product as function of temperature?
- (3) Why is the Raman spectrum similar to amorphous carbon? This seems to be inconsistent with TEM and PXRD results.
- (4) Can the authors do a simple experiment to evaluate the ORR catalytic activities of their carbon?

Reviewer #3 (Remarks to the Author):

The manuscript by Nishihara, Tani, and associates reports the successive thermal conversion of molecular organic crystals into ordered carbonaceous frameworks via crystalline intermediate COFs. The work appears to have been well done, as I ascertained by looking at the experimental details, CIF files and Movie files, in spite of precursor being previously-reported porphyrinic dimer (Ni₂-CPDPy and H₄-CPDPy). As stated by the authors, the low crystallization of H₄-CPDPy in its guest-free form causes imperfect polymerization, resulting in the collapse of its ordered structure during pyrolysis, how does it work? In my opinion, the befitting alignment of diacetylene moieties enabling solid-phase polymerization to form poly(diacetylene) backbone played an important role, but in the case of H₄-CPDPy, the close proximity (especially packing model and relative distance) of the diacetylene groups is the same as that of Ni₂-CPDPy? Are there crystallographic structural distinctions between them? Likewise, are Co₂-CPDPy/Fe₂-CPDPy isostructural with Ni₂-CPDPy, if for achieving a high HER or ORR activity? In addition, obscure statements like "these results reveal that Ni is well dispersed and exists along the structure regularity derived from the (020) plane of its precursor", and "The radicals in the dangling bonds and graphene sheets were studied by X-band electron paramagnetic resonance (EPR) analysis". Overall, the test of this material is convincing, I think many readers may be interested in this kind of materials, in view of the molecularly controlled chemical structures. The writing can be followed, I recommend the paper for publication in Nature Communications after the authors address the above points.

Reviewer #4 (Remarks to the Author):

The authors of this paper rely heavily on Rietveld refinement against powder X-ray diffraction data in order to characterize their materials. Overall, the structure determinations are well-executed and the authors do not fall into the trap of over-interpreting their results.

I have two minor issues for the authors to consider:

(i) The hydrogen atoms appear to have been refined independently, which appears unjustified by the extent and quality of the diffraction data. A riding model for the hydrogen atoms would have been more appropriate.

(ii) Did the refinement model make any use of non-crystallographic symmetry, for example in the form of restraints?

The crystallographic results and their interpretation present no barriers to publication.

Response to the reviewer #1

Thank you very much for your precious comments. We have carried out additional experiments and revised our manuscript in accordance with your opinions as shown below.

Your comment:

The authors emphasize that the resulting ordered carbonaceous frameworks exhibit an intermediate structure, between organic-based frameworks and carbon materials, with advantages pertaining to both structures. We STRONGLY request the authors to show any unique properties derived from such an intermediate structure. The authors try to characterize the obtained materials by lots of tools, I appreciate. Despite the overall high quality of the characterization, I have the negative impression that the submitted manuscript does not fulfill the impact criterium required for a publication in Nature Communications, however.

Response:

We fully agree with your comment, and carried out additional experiments. The reporting ordered carbonaceous frameworks (OCFs) have an intermediate structure, between organic-based frameworks and carbon materials. One of the structural advantages is that the molecular blocks (Ni-N₄ unit) are embedded in a conductive carbonaceous framework, and it is expected to exhibit unique electrocatalysis. We found that OCF can indeed exhibit unique electrocatalysis for selective CO₂ reduction into CO with a high Faradaic efficiency of ca. 90%. This catalysis has been reported on specific Ni cyclam complexes which have Ni-N₄ sites. However, organic complexes are not conductive and they have to be used as a thin film deposited on conductive materials such as carbon. Additionally, this selective CO₂ reduction has been rarely reported in carbonaceous materials made from Ni-based complexes. OCF shows a good electrocatalytic activity in its powder form even without any conductive additives. We have examined the catalytic activities of three reference materials to prove the uniqueness of OCF. One of them is zeolite-templated carbon (ZTC), an existing ordered microporous carbon which can be prepared by a hard template method. ZTC has the structure periodicity of 13.8 Å, which is close to those of OCFs. ZTC has an electric conductive framework which comprises mainly of *sp*² carbons, but it does not possess catalysis sites including metal species. Hence, ZTC showed no catalysis. We confirmed that Ni₂-CPD_{py} powder also does not show any catalytic activity because of its poor conductivity. Moreover, we prepared the third reference material by simple carbonization of a common Ni-based porphyrin, 5,10,15,20-tetraphenyl-21H,23H-porphine nickel(II). The resulting Ni/N/C composite is a mixture of a random-structured carbonaceous matrix and Ni metal particles. During the carbonization, Ni-N₄ units are broken, and therefore, this Ni/N/C composite also does not show any catalysis for CO₂ reduction. These results prove that the unique catalysis of OCF is derived from the Ni-N₄ units embedded in the carbonaceous framework, and such a well-defined structure cannot be achieved by carbonization of conventional Ni-complexes.

An important advantage of OCF is its good chemical designability. By means of chemistry, the structure and the chemical composition of OCF can be further varied to achieve other catalysis or function in future. We believe that aforementioned additional data demonstrate the most critical

advantage of OCF: achieving intermediate unique property of organic-based framework and carbonaceous materials.

Thus, we have added the new data of selective CO₂ electroreduction in Fig. 6, as follows.

Figure 6 Examination of selective CO₂ electro-reduction into CO. (a,b) Partial current densities used for (b) CO and (c) H₂ generation on the samples in CO₂-saturated 0.1 M KHCO₃. (c) FE for CO generation. (d) TEM image of a reference ordered microporous carbon, ZTC.

Moreover, we have thoroughly revised the section of “Electrochemical catalysis” as follows.

Compared to carbon materials, organic-based frameworks have a great advantage of chemical designability. In the latter materials, specific molecular blocks can be integrated three-dimensionally with structure order, by which a variety of unique functions can be achieved. However, they are intrinsically not electrically conductive unlike carbon, and are necessarily deposited as thin films on good conductors for electrochemical applications⁴⁸⁻⁵⁰. The proposed OCFs are expected to provide new electrocatalyst designs in which active sites are embedded in electrically conductive frameworks with structure regularity. To prove this concept, we have examined the electrocatalytic activity of the porphyrin center (Ni-N₄ unit) retained in Ni₂-CPD_{Py}973(1). Unlike the cases of Fe and Co, Ni-based complexes are generally poor in electrocatalysis for oxygen-reduction reaction. However, specific Ni cyclam complexes⁵¹⁻⁵⁴ or Ni-N-modified graphene⁵⁵, which have Ni-N₄ sites, have been reported to show unique electrocatalysis for the CO₂ reduction into CO without significant H₂ evolution and with a high Faradaic efficiency (FE) of ca. 90%. Additionally, this selective CO₂ reduction has rarely been reported in carbonaceous Ni/N/C materials made from Ni-based complexes. Fig. 6 summarizes the comparison of selective CO₂ reduction activities of Ni₂-CPD_{Py}973(1) and three reference materials, characterized by the method reported elsewhere⁵⁵. Ni₂-CPD_{Py}973(1) shows apparent CO₂ reduction catalysis into CO (Fig. 6a) even without any conductive additives, while the activity for H₂ evolution is quite low below -0.9 V vs RHE (reversible hydrogen electrode) (Fig. 6b). Thus, Ni₂-CPD_{Py}973(1) achieves high FE of 94% and 87% at -0.8 and -0.9 V, respectively

(Fig. 6c). To further investigate the uniqueness of the OCF catalysis, the same measurement was applied to three reference materials: (i) ZTC, (ii) Ni₂-CPD_{Py}, and (iii) Ni-TPP carbonized at 973 K for 1 h [Ni-TPP973(1)]. ZTC is an existing ordered microporous carbon prepared by using zeolite as a hard template², and has the structure periodicity of 13.8 Å (Fig. 6d), which is close to that of Ni₂-CPD_{Py}973(1). ZTC has an electrically conductive framework which comprises mainly of *sp*² carbons⁵⁶, but it does not possess catalysis sites including metal species. ZTC shows no catalysis in Fig. 6a, and this result clearly indicates that the ordered carbonaceous framework itself can never reduce CO₂. Ni₂-CPD_{Py} also shows no catalysis in Fig. 6a despite the presence of the Ni-N₄ unit, because the organic crystal of Ni₂-CPD_{Py} is not electrically conductive. Next, we discuss the active site. In the cases of Fe/N/C and Co/N/C electrocatalysts for oxygen reduction reaction, catalysis sites often exist as disordered forms consisting of metal species, N, and C. Hence, it is necessary to examine the catalytic activity of disordered Ni/N/C structure towards the selective CO₂ reduction into CO. For this purpose, Ni-TPP973(1) was prepared as a representative Ni/N/C material by the same carbonization procedure as that for Ni₂-CPD_{Py}973(1), from a common Ni-based porphyrin Ni-TPP. During the heat treatment, Ni-TPP is decomposed involving cleavage of Ni–N bonds, and it turns into the mixture of a disordered Ni/N/C framework and Ni metal aggregation (Supplementary Fig. 18). The resulting Ni-TPP973(1) shows no catalysis in Fig. 6a, indicating that the disordered Ni/N/C structure is not active, and the Ni-N₄ unit is the active site in Ni₂-CPD_{Py}973(1). Fig. 6 thus proves that the selective CO₂ reduction catalysis of Ni₂-CPD_{Py}973(1) can be achieved by the intermediate structure of organic-based frameworks and carbon materials, in which molecular catalysis sites (Ni-N₄) are embedded in the conductive framework. As mentioned above, conventional M/N/C catalysts have disordered structures and this has hampered the basic understanding between the structure and catalysis. With its certainly determined catalysis sites, Ni₂-CPD_{Py}973(1) can be a good platform to investigate fundamental of carbonaceous electrocatalysts.

The advantage of OCFs compared to conventional carbon materials is chemistry-based better controllability. Though the present OCFs are poorly porous (Supplementary Table 2), it is possibly improved by introducing volatile groups at designed sites of the starting molecules. Replacing Ni with other metals such as Fe, Co, Cu, Pt, and Pd can also widen the versatility of OCFs. Moreover, the exterior shape of the crystal can be also controlled based on the existing methods, for example a reprecipitation method⁵⁷⁻⁶⁰ (Supplementary Fig. 19) to achieve additional function⁵⁹. By their chemical designability, OCFs are expected to be further developed hereafter.

Response to the reviewer #2

Thank you very much for your precious comments. We have revised our manuscript in accordance with your opinions. We answer your comments one-by-one as follows:

Your comment (shown in bold):

This is a very interesting paper for preparing crystalline carbonaceous materials via pyrolysis. It can be published after consideration of the following issues:

(1) What's the role played by Ni ions? Can other metal ions (e.g., Fe) work?

Response: Thanks for your comments. Ni stabilizes porphyrin against thermochemical decomposition, and this enables to achieve a good yield of carbonization as well as the retention of the ordered structure. **To clearly explain this point, we have added the following sentence at the end of page 5.**

Ni stabilizes the porphyrin against the thermochemical decomposition, thereby achieving the better yield and retaining the ordered structure.

Moreover, we found that the Ni-N₄ units embedded in the carbonaceous framework exhibit unique catalysis for selective CO₂ reduction into CO with a high Faradaic efficiency of ca. 90%. This is also the role of Ni. **We have added the new data on the Ni-N₄ catalysis as Fig. 6, and the section of "Electrochemical catalysis" has been thoroughly revised.**

As you suggested, using other metal ions would further expand the possibility of OCFs. In the case of the present CPD_{py} matrix, Fe₂-CPD_{py} does not form crystal because Fe is trivalent (Fe³⁺) and an amorphous solid is formed by the coordination of CPD_{py} pyridyl groups to Fe. By replacing the pyridyl groups with neutral groups such as phenyl groups, the preparation of OCFs containing Fe may be possible. We are indeed trying to prepare OCFs containing other metals now. However, it significantly exceeds the scope of the present work, and we hope that we will report the results as separated papers in future. Thus, **we have added the following sentence in page 17.**

Replacing Ni with other metals such as Fe, Co, Cu, Pt, and Pd can also widen the versatility of OCFs.

(2) What's the surface area of the product as function of temperature?

Response: As described in the previous version, the OCF prepared at 873 K was found to be poorly porous. However, as you pointed out, we did not examine the effect of production temperature on the resulting surface area. Thus, we have measured the surface areas of OCFs prepared by 873 and 973 K by a N₂ adsorption technique, and the results are shown in Supplementary Table 2, as follows.

Supplementary Table 2. Porosity of Ni₂-CPD_{Py} and carbonaceous materials.

Sample	BET surface area (m ² g ⁻¹) ¹⁾	V _{N₂} (cm ³ g ⁻¹)	V _{CO₂} (cm ³ g ⁻¹)
Ni ₂ -CPD _{Py}	n.m. ^a	n.m.	0.04
Ni ₂ -CPD _{Py} 873(1)	48	0.08	0.11
Ni ₂ -CPD _{Py} 973(1)	14	0.03	n.m.

^a n.m.: Not measured.

As shown in Fig. 2g, OCF slightly shrinks from 873 to 973 K, and therefore, the porosity is decreased at a higher temperature.

The surface areas of these OCFs are 48 and 14 m² g⁻¹, respectively. As expected from their XRD patterns shown in Fig. 2g, the OCF structure shrinks at a higher temperature, resulting in the decrease of surface area. Though the present OCFs are poorly porous, this can be improved by appropriate molecular design, for example by introducing volatile groups at designed sites of the starting molecules. Thus, we have added the following sentence in page 17.

The advantage of OCFs compared to conventional carbon materials is chemistry-based better controllability. Though the present OCFs are poorly porous (Supplementary Table 2), it is possibly improved by introducing volatile groups at designed sites of the starting molecules.

We are indeed trying to synthesize porous OCFs now, and it seems to be successful. We would like to present the results by another paper in future.

(3) Why is the Raman spectrum similar to amorphous carbon? This seems to be inconsistent with TEM and PXRD results.

Response: The present OCFs retain some structure regularities of the precursor polymer crystal, but their frameworks comprise of disordered and defective graphene sheets. Though it may seem to be inconsistent, it is certainly possible. There have been several similar materials reported so far: zeolite-templated carbons and ordered mesoporous carbons (see *Adv. Mater.*, **24** (2012) 4473). In these materials, an amorphous carbon framework (found by Raman spectroscopy) forms a long-range

structure regularity which can be defined by TEM and PXRD. Thus, we have revised the last sentence in pages 10-11 as follows (changed part is shown with red colored font).

In Ni₂-CPD_{Py}873(1), most peaks disappear, and only broad D- and G-bands³⁶ originating from intervalley scattering³⁷⁻³⁹ appear, indicating that the well-defined chemical structure of the precursor polymer framework is lost and the resulting OCF consists of defective graphene sheets (Fig. 21), like zeolite-templated carbons (ZTCs) and ordered mesoporous carbons².

(4) Can the authors do a simple experiment to evaluate the ORR catalytic activities of their carbon?

Response: I think that this comment requests the data of electrocatalysis of OCFs, based on their structure advantage, *i.e.*, molecular active sites (M-N₄) are embedded in a conductive carbonaceous framework. Please note that Ni species is not active for ORR unlike Fe or Co species. Nevertheless, we have examined ORR of the present OCFs, and found they are not significantly active as expected. However, we have found the peculiar electrocatalysis of the present OCF for selective CO₂ reduction into CO with a high Faradaic efficiency of ca. 90%. This is owing to the Ni-N₄ active sites embedded in the carbonaceous framework. We have confirmed that this is unique catalysis in the present OCF, and not found in ordered microporous carbon, Ni₂-CPD_{Py} (organic crystal), and a conventional Ni/N/C composite prepared by carbonization of a common Ni complex (nickel-tetraphenylporphyrin). We have added the data of OCF electrocatalysis in the section of “Electrochemical catalysis”, together with Fig. 6.

Response to the reviewer #3

Thank you very much for your precious comments. We have revised our manuscript in accordance with your opinions. We answer your comments one-by-one as follows:

Your comment (shown in bold):

The manuscript by Nishihara, Tani, and associates reports the successive thermal conversion of molecular organic crystals into ordered carbonaceous frameworks via crystalline intermediate COFs. The work appears to have been well done, as I ascertained by looking at the experimental details, CIF files and Movie files, in spite of precursor being previously-reported porphyrinic dimer (Ni₂-CPDPy and H₄-CPDPy). As stated by the authors, the low crystallization of H₄-CPDPy in its guest-free form causes imperfect polymerization, resulting in the collapse of its ordered structure during pyrolysis, how does it work? In my opinion, the befitting alignment of diacetylene moieties enabling solid-phase polymerization to form poly(diacetylene) backbone played an important role, but in the case of H₄-CPDPy, the close proximity (especially packing model and relative distance) of the diacetylene groups is the same as that of Ni₂-CPDPy? Are there crystallographic structural distinctions between them? Likewise, are Co₂-CPDPy/Fe₂-CPDPy isostructural with Ni₂-CPDPy, if for achieving a high HER or ORR activity? In addition, obscure statements like “these results reveal that Ni is well dispersed and exists along the structure regularity derived from the (020) plane of its precursor”, and “The radicals in the dangling bonds and graphene sheets were studied by X-band electron paramagnetic resonance (EPR) analysis”. Overall, the test of this material is convincing, I think many readers may be interested in this kind of materials, in view of the molecularly controlled chemical structures. The writing can be followed, I recommend the paper for publication in Nature Communications after the authors address the above points.

(1) As stated by the authors, the low crystallization of H₄-CPDPy in its guest-free form causes imperfect polymerization, resulting in the collapse of its ordered structure during pyrolysis, how does it work? In my opinion, the befitting alignment of diacetylene moieties enabling solid-phase polymerization to form poly(diacetylene) backbone played an important role, but in the case of H₄-CPDPy, the close proximity (especially packing model and relative distance) of the diacetylene groups is the same as that of Ni₂-CPDPy? Are there crystallographic structural distinctions between them?

Response: Thanks for your comments. As you point out, well-aligned diacetylene moieties in Ni₂-CPDPy allows solid-phase polymerization, while H₄-CPDPy is not. Please note that H₄-CPDPy is not a highly crystalline solid, and its packing structure cannot be solved from the PXRD pattern (Supplementary Fig. 7). However, the presence of broad peaks in its PXRD pattern means that the solid contains irregular packing structures and distributed distances between diacetylene moieties. Thus, it is reasonably understood that H₄-CPDPy cannot be perfectly polymerized, and the original packing structure collapses during pyrolysis. To clearly explain this point, we have revised the sentences in page 8 as follows (changed part is shown with red colored font).

H₄-CPD_{py} is not a highly crystalline solid, and its packing structure cannot be solved from the PXRD pattern in its guest-free form (Supplementary Fig. 7). H₄-CPD_{py} has broad PXRD peaks, and it means that the solid contains irregular packing structures and distributed distances between diacetylene moieties, causing imperfect polymerization. Thus, the original packing structure collapses during pyrolysis (Supplementary Fig. 7).

(2) Likewise, are Co₂-CPDPy/Fe₂-CPDPy isostructural with Ni₂-CPDPy, if for achieving a high HER or ORR activity?

Response: As you suggested, using other metal ions would further expand the possibility of OCFs. In the case of the present CPD_{py} matrix, Fe₂-CPD_{py} does not form crystal because Fe is trivalent (Fe³⁺) and an amorphous solid is formed by the coordination of CPD_{py} pyridyl groups to Fe. By replacing the pyridyl groups with neutral groups such as phenyl groups, the preparation of OCFs containing Fe may be possible. We are indeed trying to prepare OCFs containing other metals now. However, it significantly exceeds the scope of the present work, and we hope that we will report the results as separated papers in future. Thus, we have added the following sentence in page 17.

Replacing Ni with other metals such as Fe, Co, Cu, Pt, and Pd can also widen the versatility of OCFs.

On the other hand, you may mean that a unique function based on the OCFs structure should be shown. As you know, Ni species is not active for ORR unlike Fe or Co species, and it would be the reason why you are interested in the electrocatalysis of Co₂-CPD_{py}/Fe₂-CPD_{py}. We have actually found the peculiar electrocatalysis of the present Ni-based OCFs for selective CO₂ reduction into CO with a high Faradaic efficiency of ca. 90%. This is owing to the Ni-N₄ active sites embedded in the carbonaceous framework. We have confirmed that this is unique catalysis in the present OCF, and not found in ordered microporous carbons, Ni₂-CPD_{py} (organic crystal), and a conventional Ni/N/C composite prepared by carbonization of a common Ni complex (nickel-tetraphenylporphyrin). We have added the data of OCF electrocatalysis in the section of “Electrochemical catalysis”, together with Fig. 6.

(3) In addition, obscure statements like “these results reveal that Ni is well dispersed and exists along the structure regularity derived from the (020) plane of its precursor”, and “The radicals in the dangling bonds and graphene sheets were studied by X-band electron paramagnetic resonance (EPR) analysis”.

Response: Thank you very much for your comments. We have revised these sentences as follows (changed part is shown with red colored font).

The results of TEM, HAADF-STEM, and synchrotron PXRD reveal that Ni is not aggregated as Ni metal or NiO, and exists along the structure regularity derived from the (020) plane of its

precursor. As shown later, Ni retains its original coordination structure (Ni-N₄) in the carbonaceous framework.

The microstructures of OCFs were further studied by X-band electron paramagnetic resonance (EPR) analysis (Supplementary Fig. 14). Ni₂-CPD_{Py}873(1) at 5.0 K displayed an absorption line ($g \sim 2$) with a sharp line-width (2.8 mT). These results indicate the formation of organic radicals derived from dangling bonds generated during the carbonization process.

Response to the reviewer #4

Thank you very much for your precious comments. We have revised our manuscript in accordance with your opinions. We answer your comments one-by-one as follows:

Your comment (shown in bold):

The authors of this paper rely heavily on Rietveld refinement against powder X-ray diffraction data in order to characterize their materials. Overall, the structure determinations are well-executed and the authors do not fall into the trap of over-interpreting their results.

I have two minor issues for the authors to consider:

(1) The hydrogen atoms appear to have been refined independently, which appears unjustified by the extent and quality of the diffraction data. A riding model for the hydrogen atoms would have been more appropriate.

Response: Thanks for your comments. The software we used (PDXL) is not equipped with a structure refinement tool using a riding model, but it is possible to obtain almost the same result by properly using an alternative restraint function. We have optimized the restraint parameters to achieve the similar result to that obtained by the riding model. Hydrogen atoms very little contribute to the PXRD results, and their positions can be refined almost automatically by the restraint function, after the determination of the positions of the non-hydrogen atoms. Hence, the result becomes almost the same as that obtained by the riding model. We have added the following sentences in the “PXRD analysis” section in Supplementary Information.

Since PDXL is not equipped with a structure refinement tool using a riding model, the positions of hydrogen atoms were determined by the restraint function in which the restraint parameters were optimized to achieve the similar result to that obtained by the riding model. Hydrogen atoms very little contribute to the PXRD results, and their positions can be refined almost automatically by the restraint function, after the determination of the positions of the non-hydrogen atoms. Hence, the result becomes almost the same as that by the riding model.

(2) Did the refinement model make any use of non-crystallographic symmetry, for example in the form of restraints?

Response: Thanks for your comments. We have used restraint, but the parameters are not taken from non-crystallographic symmetry. The restraint parameters were taken from the Cambridge Crystallographic Data Centre (CCDC) by using the Mogul software. Therefore, the parameters are based on the existing crystallographic symmetry. Thus, we have added the following sentences in the “PXRD analysis” section in Supplementary Information.

The structure refinement was carried out with a restraint function available in PDXL. The restraint parameters were taken from the Cambridge Crystallographic Data Centre (CCDC) by using the Mogul software.

REVIEWERS' COMMENTS:

Reviewer #1 (Remarks to the Author):

The revised manuscript is really improved. I can accept the publication of this paper.

Reviewer #2 (Remarks to the Author):

The revised paper addressed the issues raised and can be published.

Reviewer #3 (Remarks to the Author):

I read the revised manuscript and response letter carefully. The authors had solved the most of issues commented by the reviewers. I think authors' response on my comments is appropriate, therefore, I suggest that the paper be accepted in Nat. Commun.